**Data Availability Statement:** Data cannot be shared publicly because of Kyoto University. Data

# Energy landscape analysis and time-series clustering analysis of patient state multistability related to rheumatoid arthritis drug treatment: The KURAMA cohort study

**Keiichi Yamamoto**[1]*, **Masahiko Sakaguchi**[2], **Akira Onishi**[3], **Shinichiro Yokoyama**[4], **Yusuke Matsui**[4], **Wataru Yamamoto**[3,5], **Hideo Onizawa**[3], **Takayuki Fujii**[3], **Koichi Murata**[3], **Masao Tanaka**[3], **Motomu Hashimoto**[6], **Shuichi Matsuda**[3], **Akio Morinobu**[3]

1 Division of Data Science, Center for Industrial Research and Innovation, Translational Research Institute for Medical Innovation, Osaka Dental University, Hirakata City, Osaka, Japan, 2 Department of Engineering Informatics, Faculty of Information and Communication Engineering, Osaka Electro-Communication University, Neyagawa City, Osaka, Japan, 3 Department of Advanced Medicine for Rheumatic Diseases, Kyoto University Graduate School of Medicine, Sakyo, Kyoto, Japan, 4 Oracle, Tokyo, Japan, 5 Department of Health Information Management, Kurashiki Sweet Hospital, Nakasho, Kurashiki, Kurashiki City, Okayama Prefecture, Japan, 6 Department of Clinical Immunology, Osaka Metropolitan University Graduate School of Medicine, Osaka City, Japan

* yamamoto-k@cc.osaka-dent.ac.jp

## Abstract

Rheumatoid arthritis causes joint inflammation due to immune abnormalities, resulting in joint pain and swelling. In recent years, there have been considerable advancements in the treatment of this disease. However, only approximately 60% of patients achieve remission. Patients with multifactorial diseases shift between states from day to day. Patients may remain in a good or poor state with few or no transitions, or they may switch between states frequently. The visualization of time-dependent state transitions, based on the evaluation axis of stable/unstable states, may provide useful information for achieving rheumatoid arthritis treatment goals. Energy landscape analysis can be used to quantitatively determine the stability/instability of each state in terms of energy. Time-series clustering is another method used to classify transitions into different groups to identify potential patterns within a time-series dataset. The objective of this study was to utilize energy landscape analysis and time-series clustering to evaluate multidimensional time-series data in terms of multistability. We profiled each patient's state transitions during treatment using energy landscape analysis and time-series clustering. Energy landscape analysis divided state transitions into two patterns: "good stability leading to remission" and "poor stability leading to treatment dead-end." The number of patients whose disease status improved increased markedly until approximately 6 months after treatment initiation and then plateaued after 1 year. Time-series clustering grouped patients into three clusters: "toward good stability," "toward poor stability," and "unstable." Patients in the "unstable" cluster are considered to have clinical courses that are difficult to predict; therefore, these patients should be treated with more care. Early disease detection and treatment initiation are important. The evaluation of state multistability enables us to understand a patient's current state in the context of overall state

are available from the Kyoto University Institutional Data Access / Ethics Committee (contact via Kyoto University) for researchers who meet the criteria for access to confidential data.

**Funding:** This research was funded by a Grant-in-Aid for Scientific Research (19K12867). The funder played no role in study design, data collection, analysis and interpretation of data, or the writing of this manuscript.

**Competing interests:** he Department of Advanced Medicine for Rheumatic Diseases, Kyoto University Graduate School of Medicine, is supported by Nagahama City, Shiga, Japan; Toyooka City, Hyogo, Japan; Asahi Kasei Pharma Corp.; and AYUMI Pharmaceutical Co. MH received research grants and/or speaker fees from Abbvie, Asahi Kasei, Astellas, Ayumi, Brystol Meyers, Chugai, EA Pharma, Eisai, Daiichi Sankyo, Eli Lilly, Novartis Pharma, and Tanabe Mitsubishi. M.T. received research grants and/or speaker fees from AbbVie GK, Asahi Kasei Pharma Corp., Astellas Pharma Inc., Chugai Pharmaceutical Co., Ltd., Daiichi Sankyo Co., Ltd., Eisai Co., Ltd., Eli Lilly Japan K.K., Janssen Pharmaceutical K.K., Kyowa Kirin Co., Ltd., Pfizer Inc., Taisho Pharmaceutical Co., Ltd., Tanabe Mitsubishi Pharma Corp., Teijin Pharma, Ltd., and UCB Japan Co., Ltd. K.M. received speaking and/or consulting fees from AbbVie GK, Eisai Co., Ltd., Pfizer Inc., Chugai Pharmaceutical Co., Ltd., Mitsubishi Tanabe Pharma Corp., Bristol-Myers Squibb, Daiichi Sankyo Co., Ltd., Janssen Pharmaceutical K.K., and Asahi Kasei Pharma Corp. H.O. has received speaker fees from AbbVie, Asahi Kasei, Astellas Pharma Inc., Eisai Co. Ltd., Janssen Pharmaceutical K.K., Mitsubishi Tanabe Pharma Corporation, and Daiichi Sankyo Co. Ltd. The other authors declare that they have no conflicts of interest. This does not alter our adherence to PLOS ONE policies on sharing data and materials.

transitions related to rheumatoid arthritis drug treatment and to predict future state transitions.

## Introduction

Rheumatoid arthritis (RA) causes joint inflammation due to immune abnormalities, resulting in joint pain and swelling. In recent years, there have been considerable advancements in the treatment of RA, partly due to the development of drugs such as methotrexate (MTX), biologic disease-modifying anti-rheumatic drugs (bDMARDs), and targeted synthetic DMARDs (tsDMARDs) such as Janus kinase (JAK) inhibitors; furthermore, the "treat-to-target (T2T) algorithm", in which treatment is periodically adjusted to a target disease index, has led to improvements in RA treatment [1–5]. However, even with these approaches, only approximately 60% of patients achieve remission. Therefore, 10–20% of RA patients who are treatment-refractory have been identified as having difficult-to-treat (D2T) RA. In 2020, the European League Against Rheumatism (EULAR) published the D2T RA EULAR Definition. The development of appropriate treatments for refractory patients is urgently needed [6–8].

Due to advances in information technology, a variety of digitized data from daily practice, such as electronic medical records and various laboratory test values, can be collected. Therefore, expectations regarding "real-world evidence" are increasing [9–11]. However, the aim of many clinical studies is to consider the whole treatment as a "single intervention" and to determine the effectiveness and safety of that intervention while eliminating bias as much as possible. Multidimensional time-series data over the entire course of treatment are rarely analyzed in a temporal manner. In contrast, daily practice requires that the patient's condition be observed over time and that the treatment method be selected in a timely manner to optimize efficacy and survival time. In other words, in daily practice, the whole treatment is not considered a "single intervention." Instead, the optimal treatment is selected based on the constantly changing state of the patient. To obtain high-quality real-world evidence, it is necessary to develop a method for analyzing multidimensional time-series data in a time-dependent manner over the course of treatment and providing information that is more consistent with decision-making in daily practice [12, 13].

In multidimensional time-series data analysis in medicine, there are examples of applying dynamic treatment regimens (DTRs) and deep learning. However, establishing DTRs requires data on all treatments beginning at the initial visit in addition to an intermediate covariate history. The computational requirements for model building are high. Therefore, DTRs are often used in clinical trials or observational studies where multiple treatment regimens that are predicted in advance are compared in a time-dependent manner [14–17]. Furthermore, the logic of deep learning is not very transparent. In many cases, the rationale is unclear, thus making it difficult to implement deep learning in daily practice [18–24].

The Kyoto University Rheumatoid Arthritis Management Alliance (KURAMA) cohort database of the Rheumatology Center of Kyoto University Hospital includes data on initial consultation, follow-up, blood tests, etc., from all patients with RA visiting the center. The database has both clinical and research applications. It includes information on more than 3,000 patient registrations with more than 40,000 disease activity data points and has already been used extensively for the analysis of drug administration and disease activity [25–28]. The KURAMA cohort database includes high-quality, multidimensional time-series data and is considered to be suitable for time-series data analysis.

Energy landscape analysis is used to estimate a model distribution of maximum likelihood from data, making it possible to visualize the ease with which transitions occur between high- and low-energy states and thus enabling intuitive interpretation [29, 30]. Patients with chronic or multifactorial diseases shift between states from day to day. These patients may remain in a good or poor state with little change, or their condition may vary frequently. Energy landscape analysis can quantitatively evaluate the stability/instability of each state as energy, which is not possible with the conventional method of simply clustering state variables. By assessing stable/unstable states in addition to conventional good/poor states, energy landscape analysis makes it possible to determine when interventions are effective and which states are difficult to treat. Energy landscape analysis is often used in areas such as protein folding and stability analysis [31–33] and is considered useful for visualizing the state transitions of patients in real-world practice.

Time-series clustering is another method used to classify transitions into different groups to identify potential patterns within a time-series dataset [34–39]. Dynamic time warping (DTW) is a type of time-series clustering that measures the distance and similarity between time-series data by finding the shortest path, which can be identified by summing the distance (i.e., the absolute value of the error) between the points of two time series. DTW can determine the similarity of time series even if the length and period of the time series are different [35, 40].

We hypothesized that using energy landscape analysis and time-series clustering with DTW to evaluate the multidimensional time-series data in the KURAMA cohort would enable the visualization of transitions between time-dependent states in patients with RA during drug treatment, thus providing useful information for achieving the goals of RA treatment.

The purpose of this study was to utilize energy landscape analysis and time-series clustering with DTW to evaluate multidimensional time-series data in the KURAMA cohort in terms of multistability, thereby facilitating the achievement of RA treatment goals.

## Materials and methods

### Study design and participants

This single-center, retrospective, observational study utilized the KURAMA cohort database (S1 Checklist). The study protocol adhered to the principles of the Declaration of Helsinki and was approved by the Kyoto University Graduate School of Medicine Medical Ethics Committee through a central collective review (R2820), and written informed consent for study participation was obtained from all patients. The study received approval on March 30, 2021, granting access to the data from that point onward.

All patients who presented to the Rheumatology Center of Kyoto University Hospital and who met the 1987 or 2010 RA classification criteria [41] were enrolled in the KURAMA cohort study, and clinical and functional data were recorded at baseline and at each visit during the study. The inclusion criteria were as follows: patients with RA enrolled between January 1, 2011, and December 31, 2018; no previous medication history at the first visit; and onset of clinical remission within 3 years or follow-up for up to 3 years.

### Variables

We defined a model of patient state transitions in RA drug treatment based on T2T. The patient's state shifts from time to time. Fig 1 shows the ease and direction of transition of the patient's state based on high and low energy in RA practice.

When energy is low, the patient is stable in a good or poor state. On the other hand, when the energy is high, the patient is unstable and transitions easily to another state. The high-energy state is considered to indicate the period of effective treatment. Individual states were

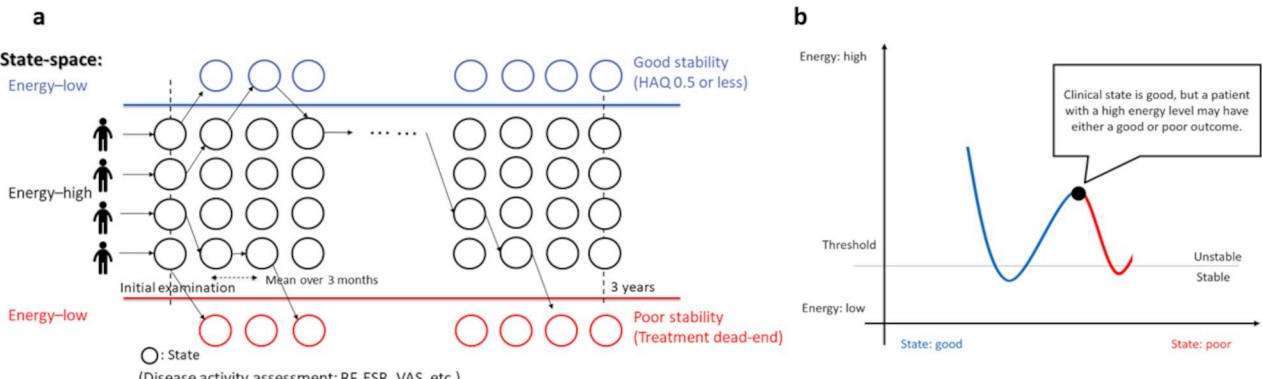

**Fig 1. Patient state transition model in RA drug treatment.** The higher the energy is, the more unstable the state and the easier it is to obtain a therapeutic effect. At initial consultation, many patients present in a poor state and with high disease activity (i.e., high energy). With treatment, they may transition to a good state or stabilize in a poor state. The physician observes the patient's state and executes a treatment strategy to stabilize the patient in a good state or to prevent stabilization in a poor state below a certain energy threshold. The objective of this study was to visualize the state transitions of these patients as a population.

evaluated using a comprehensive disease activity assessment. "Good stability" was defined as meeting functional remission criteria based on the Health Assessment Questionnaire (HAQ), and "poor stability" was defined as falling below the energy threshold without functional remission (i.e., treatment dead-end) [42]. The term "treatment dead-end" indicates that the patient will be not in remission, regardless of all potential future treatment sequences. In this study, the treatment period was 3 years, and the state of each patient was evaluated up to 12 times. In RA practice, the physician collects information about the patient's state through visual examination, palpation, blood tests, and imaging tests. Variables related to disease activity, bone destruction, and immunological response are used in comprehensive disease activity assessments in daily practice and were used to characterize the patient's state in this study, including:

- Rheumatoid factor (RF)

- Erythrocyte sedimentation rate at 1 hour (ESR1h)

- Patient's visual analog scale (PtVAS)

- Doctor's visual analog scale (DrVAS)

- State of bone destruction according to Steinbrocker's staging classification (STAGE) [43]

- Swollen joint count– 28 joints (SJC28)

- Tender joint count– 28 joints (TJC28)

The time-series data that served as input values for the energy landscape analysis were binarized as high activity (i.e., nonremission) = 1 and low activity (i.e., remission) = -1. RF and ESR1h were binarized based on blood test reference values. PtVAS, DrVAS, SJC28, and TJC28 were binarized based on Boolean remission criteria [44], and STAGE was binarized based on Steinbrocker's staging classification [43]. The remission criteria were as follows:

- RF = 15 #Unit: IU/mL

- ESR1h (male) = 10 #units: mm

- ESR1h (female) = 20 #units: mm

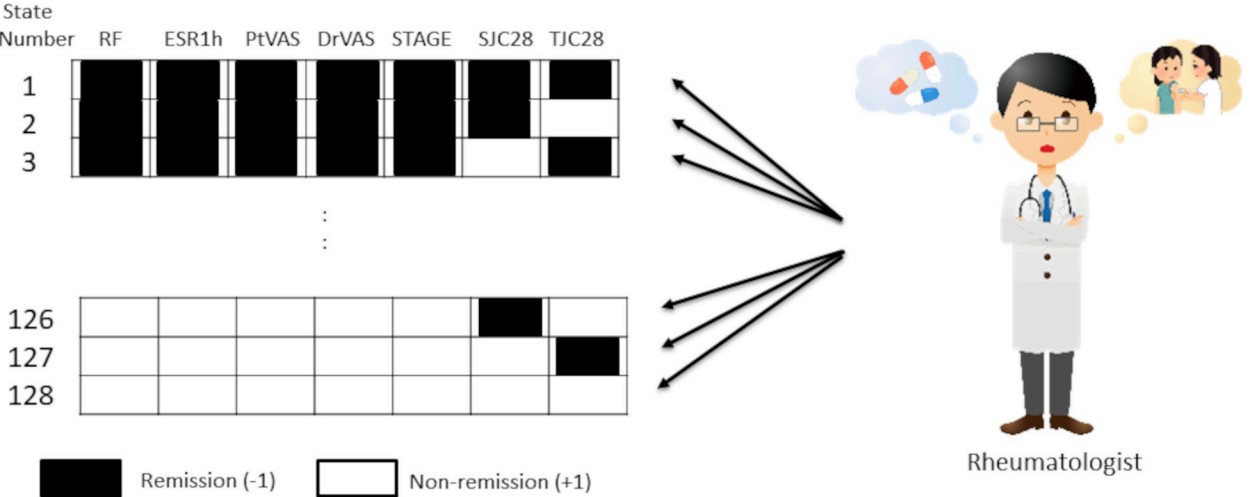

**Fig 2. Treatment selection by the Rheumatologist based on state number.** Physicians select the optimal treatment according to seven test results (i.e., RF, ESR1h, PtVAS, DrVAS, STAGE, SJC28, and TJC28) that are used in comprehensive disease activity assessments in daily practice.

- PtVAS = 10 #100mm VAS

- DrVAS = 10 #100mm VAS

- STAGE = 3 #stage 1, 2 or stage 3, 4

- SJC28 = 1

- TJC28 = 1

Since there are two patterns per factor (i.e., +1/-1), analyzing seven factors generates 128 (i.e., 2^7) activity patterns. The physician selected the optimal treatment according to this information (i.e., state number) (Fig 2).

Medications for RA were reviewed at least every 3 months until treatment goals were achieved according to T2T. The drug therapies were categorized as follows: "conventional synthetic DMARDs (csDMARDs) including MTX"; bDMARDs, including "cytotoxic T lymphocyte antigen 4-immunoglobulin G (CD80/86)", "interleukin-6 (IL6) inhibitor," "tumor necrosis factor (TNF) inhibitor"; and "tsDMARDs (JAK)".

### Analytical methods

**Overview of the analytical methods.** Energy landscape analysis was used to assess and visualize state multistability as "energy." Dynamic systems were formulated with a Boltzmann machine that used the Boltzmann distribution with energy to define the probability distribution. The Boltzmann machine reduced multidimensional data to a measurement considered to represent "energy." An index of multistability was assigned to each state to identify whether the disease condition was improving or worsening. Furthermore, multidimensional time-series clustering of patients enabled us to classify and visualize the time-series transitions of unstable and stable states.

**Energy landscape analysis and the Boltzmann machine.** For each patient ($s$), there are seven different factor values ($i$) observed at each of the 12 time points ($t$) every 3 months over 3 years. Let $\Omega = \{TJC28, SJC28, STAGE, DrVAS, PtVAS, ESR1h, RF\}$ and $T = \{1,2,\ldots,11,12\}$. Each factor $x_i(t)$ is binarized (-1 or +1) according to the clinical criteria for $i \in \Omega$ and $t \in T$.

We apply a Boltzmann machine [45] that extends the deterministic dynamics of the Hopfield model, a well-known model of associative memory, to stochastic dynamics.

The Boltzmann machine is defined as a multidimensional Boltzmann distribution. The definition of the distribution includes energy. A stochastic model on an undirected graph G ($\Omega$, E) is defined, where E is the set of ($i,j$) links between nodes $i,j \in \Omega$.

$$P(\mathbf{X} = (x_1, x_2, \ldots, x_6, x_7)|\boldsymbol{\theta}, \mathbf{W}) \stackrel{\text{def}}{=} \frac{\exp(-\Phi(\mathbf{X} = (\mathbf{x}_1, \mathbf{x}_2, \ldots, \mathbf{x}_6, \mathbf{x}_7)|\boldsymbol{\theta}, \mathbf{W}))}{Z(\boldsymbol{\theta}, \mathbf{W})}$$

$$Z(\boldsymbol{\theta}, \mathbf{W}) \stackrel{\text{def}}{=} \sum_{\mathbf{x}_i = -1, +1, i \in \Omega} \exp(-\Phi(\mathbf{X} = (\mathbf{x}_1, \mathbf{x}_2, \ldots, \mathbf{x}_6, \mathbf{x}_7)|\boldsymbol{\theta}, \mathbf{W}))$$

$$\Phi(\mathbf{X} = (x_1, x_2, \ldots, x_6, x_7)|\boldsymbol{\theta}, \mathbf{W}) \stackrel{\text{def}}{=} -\sum_{i \in \Omega} \theta_i \mathbf{x}_i - \sum_{i,j \in \Omega} \mathbf{w}_{ij} \mathbf{x}_i \mathbf{x}_j,$$

where $x_i = -1$ or $+1$, vector $\boldsymbol{\theta} = \{\theta_i\}$, and matrix $\mathbf{W} = \{w_{ij}\}$ for $i,j \in \Omega$. G ($\Omega$, E) are assumed in the link structure of the Boltzmann machine, such that $w_{ij} = w_{ji}$ and $w_{ii} = 0$. The third equation is called the energy function. The second equation is the normalizing constant of the first equation. According to the definition of the first equation, this Boltzmann machine is a mathematical model in which the energy function is smaller than the monotonically increasing nature of the exponential function, and the activity pattern appears with higher probability.

The parameters $\boldsymbol{\theta}$ and $\mathbf{W}$ are trained to match the probability of occurrence of the actual observed data. For each person $s \in S$, the observation vector obtained at each time $t \in T$ is assumed to be generated from an independent homoscedastic distribution. For the obtained dataset D = $\{x_i^s(t)|i \in \Omega, t \in \mathrm{T}, s \in S\}$, the maximum likelihood method is applied to estimate the parameters s $\boldsymbol{\theta}$ and $\mathbf{W}$ that satisfy the following equation:

$$(\boldsymbol{\theta}, \mathbf{W}) = \operatorname*{argmax}_{\boldsymbol{\theta}, \mathbf{W}} L_D(\boldsymbol{\theta}, \mathbf{W})$$

$$L_D(\boldsymbol{\theta}, \mathbf{W}) \stackrel{\text{def}}{=} \prod_{t \in \mathrm{T}, s \in \mathrm{S}} \mathsf{P}(\mathbf{X} = (\mathbf{x}_1^s(t), \mathbf{x}_2^s(t), \ldots, \mathbf{x}_6^s(t), \mathbf{x}_7^s(t))|\boldsymbol{\theta}, \mathbf{W})$$

The parameter that maximizes this log-likelihood function, log $L_D(\boldsymbol{\theta}, \mathbf{W})$, is obtained using the gradient ascent method as follows:

$$\theta_i^{new} - \theta_i^{old} = \varepsilon \left\{ \frac{\sum_{t \in \mathrm{T}, s \in \mathrm{S}} x_i^s(t)}{12|S|} - E_{old}[X(i)] \right\},$$

$$J_{ij}^{new} - J_{ij}^{old} = \varepsilon \left\{ \frac{\sum_{t \in \mathrm{T}, s \in \mathrm{S}} x_i^s(t) x_j^s(t)}{12|S|} - E_{old}[X(i)X(j)] \right\},$$

where $E_{old}$ is the expected value from the Boltzmann distribution using the previous parameter in the dataset, $X(i)$ is the $i$-th element of $X,i,j \in \Omega$, $|S|$ is the number of elements in S, and $\varepsilon = 0.2$ is the learning rate. Up to 5,000,000 iterations were performed.

**Disconnectivity graph.** First, a minimal activity pattern $X = \{x_i|i \in \Omega\}$ is defined such that for any activity pattern $Y$ with one different node in $X$, $\Phi(Y) \geqq \Phi(X)$ holds for the given parameters s $\boldsymbol{\theta}$ and $\mathbf{W}$. Next, a path $a$ with $X$ and $Y$ is defined. Let A($X,Y,a$) = $\{Z|Z$ transform $X$ one node at a time until it can be transformed into $Y\}$. Note that this set exists for the number of paths $a$. Finally, the energy of the highest hill to be surmounted by path $a$ is $\max_{A(X,Y,a)} \Phi(Z)$.

The energy barrier of the transition from $X$ to $Y$ is defined as $\min_a[\max_{A(X,Y,a)} \Phi(Z)] - \Phi(X)$ [29, 46, 47].

**Time-series clustering.** We chose the variables $y^{id}(t)$ based on the activity pattern and chose the energy and HAQ for patient $id$ and time $t$. Clustering was performed using the k-means method with distances based on the dynamic contraction method toward time-series vectors $\{y^{id}(t)\}_{t \in T}$, where $y^{id}(t) \in \mathbb{R}^{|\Omega|} \times \mathbb{R}^2$, for $id = 1,2,\cdots,$N, T is the time point, and N is the number of samples [48].

The number of clusters K was determined by the elbow method or silhouette analysis [49, 50]. The initial central value $\{c^{id}(t)\}_{t \in T}$ of cluster k was randomly chosen for $c^{id}(t) \in \mathbb{R}^{|\Omega|} \times \mathbb{R}^2$. The following method of minimizing the within-cluster sum of squared errors of prediction (SSE) was used. Iterations by the k-means method were performed up to 50 times. The center value was updated by the mean vector in the cluster.

$$\text{SSE} = \sum_{k=1}^{K} \sum_{id=1}^{N} \delta_{(id,k)} \{DTW(\{y^{id}(t)\}_{t \in T}, \{c^{id}(t)\}_{t \in T})\}^2$$

$$\delta_{(id,k)} = \begin{cases} 1, & if \ id = \underset{k \in \{1,2,\cdots,K\}}{\text{argmin}} \ DTW(\{y^{id}(t)\}_{t \in T}, \{c^{id}(t)\}_{t \in T}) \\ 0, & otherwise \end{cases}$$

The distance function DTW: $\mathbb{R}^u \times \mathbb{R}^v \to \mathbb{R}$ of two time-series vectors $\{x_t\}_{t=u}^{S}, \{y_t\}_{t=v}^{S}$ is defined as follows:

$$DTW(\{x_S, \{y_t\}_{t=v}^{S}) = \sum_{t=v}^{S} |x_S - y_t|, \ DTW(\{x_t\}_{t=u}^{S}, y_S) = \sum_{t=u}^{S} |x_t - y_S|,$$

$$\begin{aligned} DTW(\{x_t\}_{t=u}^{S}, \{y_t\}_{t=v}^{S}) \\ = |x_u - y_v| \\ + \min[DTW(\{x_t\}_{t=u+1}^{S}, \{y_t\}_{t=v}^{S}), DTW(\{x_t\}_{t=u+1}^{S}, \{y_t\}_{t=v+1}^{S}), DTW(\{x_t\}_{t=u}^{S}, \{y_t\}_{t=v+1}^{S})], \end{aligned}$$

for u<S and v<S, where $|\cdot|$ is the Euclidean distance. The DTW algorithm [51] for calculating the distance between two time series uses least-cost elasticity matching, which does not allow the time series to intersect.

## Analysis environment and preprocessing

Oracle Autonomous Data Warehouse and Oracle Cloud Infrastructure Data Science were used as analysis environments. The Statistics and Machine Learning Toolbox and the Energy Landscape Analysis Toolbox v1.2 [29] of MATLAB R2022b were used for energy landscape analysis.

For missing value completion, STAGE, which comprises categorical data, was substituted for the before and after data. All other factors were numerical data, and linear interpolation of time-series data was used; for patients with fewer than 12 points, data were generated in the same way as for missing value imputation above. For example, for patients who entered remission or who withdrew from the study, data were generated in the same way as for the final point until the patients reached 3 years of remission. In the case of missing time-series points, categorical STAGE data were assigned before and after the values, and the numerical data were linearly interpolated.

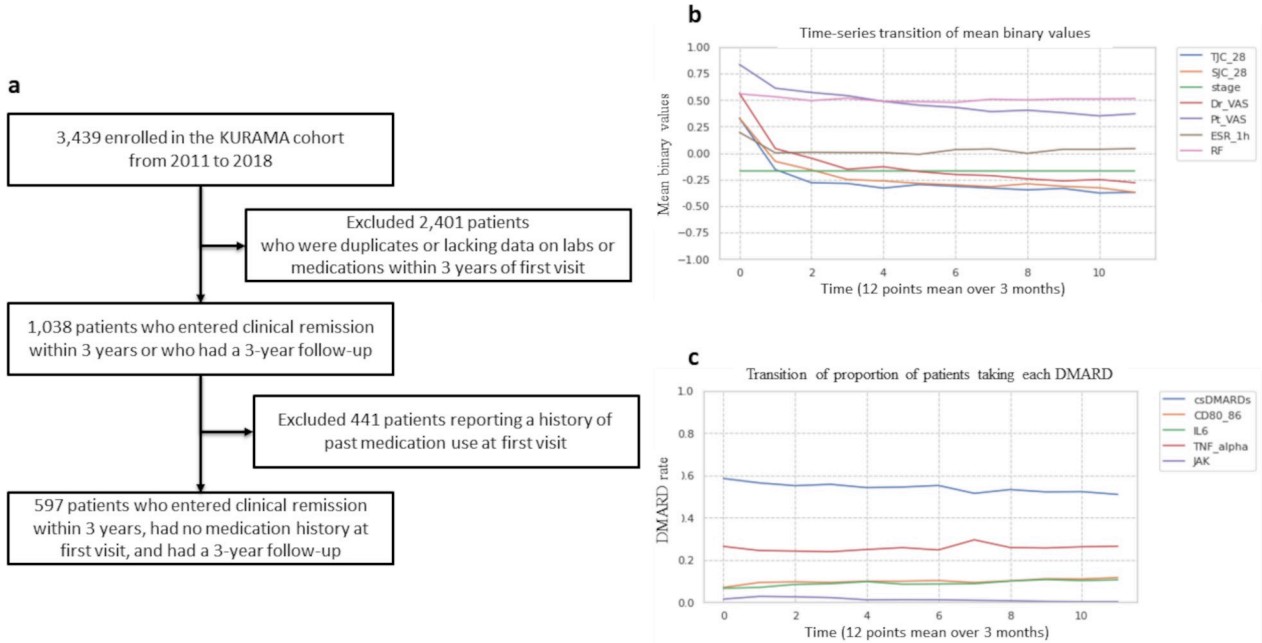

**Fig 3. Subject selection flowchart, binarized averages of the seven factors, and distributions of administered drug therapies.** (A) The subject selection flowchart. (B) Binarized averages of the seven factors over 3 years. (C) 3-year trends in the distributions of administered drug therapies. The horizontal axis shows the number of points (0–11) at which the treatment effect was assessed (every 3 months). The 3-year average values of the seven factors were as follows: DrVAS = -0.11, Pt VAS = 0.49, TJC28 = -0.26, SJC28 = -0.22, STAGE = -0.17, RF = 0.51, and ESR1h = 0.03. Regarding the percentages of patients receiving each drug therapy, the 3-year averages were as follows: csDMARDs = 54%, CD80/86 = 10%, IL6 = 9%, TNFα = 26%, and JAK = 1%.

## Results

### Participants and descriptive data

A flowchart of the subject selection, the 3-year trends for the seven factor values, and the distribution of the drug therapies administered are shown in Fig 3.

From the 3,439 individuals enrolled in the KURAMA cohort from 2011 to 2018, we excluded 2,401 individuals who were duplicates or who lacked laboratory values or medication information within 3 years of their first visit. In addition, we excluded 441 patients with a history of past medication use at the time of their first visit. Thus, 597 patients were ultimately included in the analysis.

The baseline characteristics of the 597 subjects are shown in Table 1.

### Results of the energy landscape analysis

The parameters of the Boltzmann machine were as follows:

$\theta$ = -0.4259 –0.1940 –0.2995 –0.0313 0.7974 –0.0784 0.6501

$W$ = 0 0.1922 0.0028 0.5739 0.2281 –0.0108 0.0189

0.1922 0 0.2048 0.5580 0.0284 0.1376 –0.0332

0.0028 0.2048 0 0.1392 0.1422 0.0141 0.1811

0.5739 0.5580 0.1392 0 0.2834 0.1383 0.0793

0.2281 0.0284 0.1422 0.2834 0 0.0850 –0.0103

-0.0108 0.1376 0.0141 0.1383 0.0850 0 0.2284

0.0189 –0.0332 0.1811 0.0793 –0.0103 0.2284 0

**Table 1. Baseline characteristics of the 597 subjects.**

| Sex | | |
|---|---|---|
| | Male, n (%) | 123 (20.6%) |
| | Female, n (%) | 474 (79.4%) |
| Age, laboratory values at initial examination, etc. | | |
| (Mean±SD, Median (Range)) | | |
| | Age at initial examination (years) | 61.1 ± 13.1, 63.0 (17.0–91.0) |
| | Duration of illness (years) | 9.2 ± 11.0, 4.9 (0.1–62.4) |
| | Age at disease onset (years) | 51.9 ± 15.0, 53.0 (5.0–89.0) |
| | HAQ | 0.7 ± 0.7, 0.5 (0.0–3.0) |
| | mHAQ | 0.5 ± 0.6, 0.4 (0.0–2.9) |
| | ACPA (U/mL) | 175.1 ± 255.7, 76.9 (0.5–1960.0) |
| | RF (IU/mL) | 138.6 ± 250.7, 49.2 (6.0–2173.8) |
| | CRP (mg/dL) | 1.4 ± 2.4, 0.4 (0.0–23.4) |
| | ESR (mm/h) | 32.8 ± 25.9, 24.0 (0.0–124.0) |
| | CDAI | 14.5 ± 10.8, 12.0 (0.0–66.0) |
| | SDAI | 16.3 ± 12.3, 13.6 (0.2–71.8) |
| | DAS28-ESR | 4.1 ± 1.5, 4.1 (1.0–8.1) |
| | DAS28-CRP | 3.5 ± 1.4, 3.4 (1.0–7.8) |
| Stage, Class | | |
| | Stage 1, 2, 3, 4 | 33.5%, 25.0%, 14.7%, 26.8% |
| | Class 1, 2, 3, 4 | 20.9%, 63.1%, 14.9%, 1.0% |

Energy landscape analysis (Fig 4, Table 2, and S1 Table) revealed that patient state could be divided into two fixed disease state patterns: "good stability leading to remission" (blue) and "poor stability (i.e., treatment dead-end) (red).

Fig 4A shows the activity patterns. The numbers shown in the directed graph in Fig 4A are unique state numbers. The scale to the right indicates the high and low energy values of each state. The higher the energy value is, the more unstable the state is and the easier it is to transition to another state. On the other hand, the lower the energy value is, the more fixed the state is and the harder it is to transition to another state. In other words, higher energy values indicate that the patient is more responsive to drug treatment or that the rate of disease deterioration exceeds the effect of drug treatment. On the other hand, lower energy levels mean that the patient is more likely to remain in the current state and that the disease is less likely to improve or worsen.

Fig 4B shows whether each of the seven factors met the remission and nonremission criteria according to the state numbers that take the minimal energy of the patterns of good stability and poor stability. The factors shown in black met the remission criteria, while those in white did not. In the "good stability" pattern, all factors except RF and PtVAS met the remission criteria. On the other hand, in the "poor stability" pattern, none of the factors reached the remission criteria.

The patient's state could switch between the two patterns as a result of drug treatment effects or other factors. Fig 4C shows the threshold value of the energy at which pattern transitions can occur (see: "*Disconnectivity graph*").

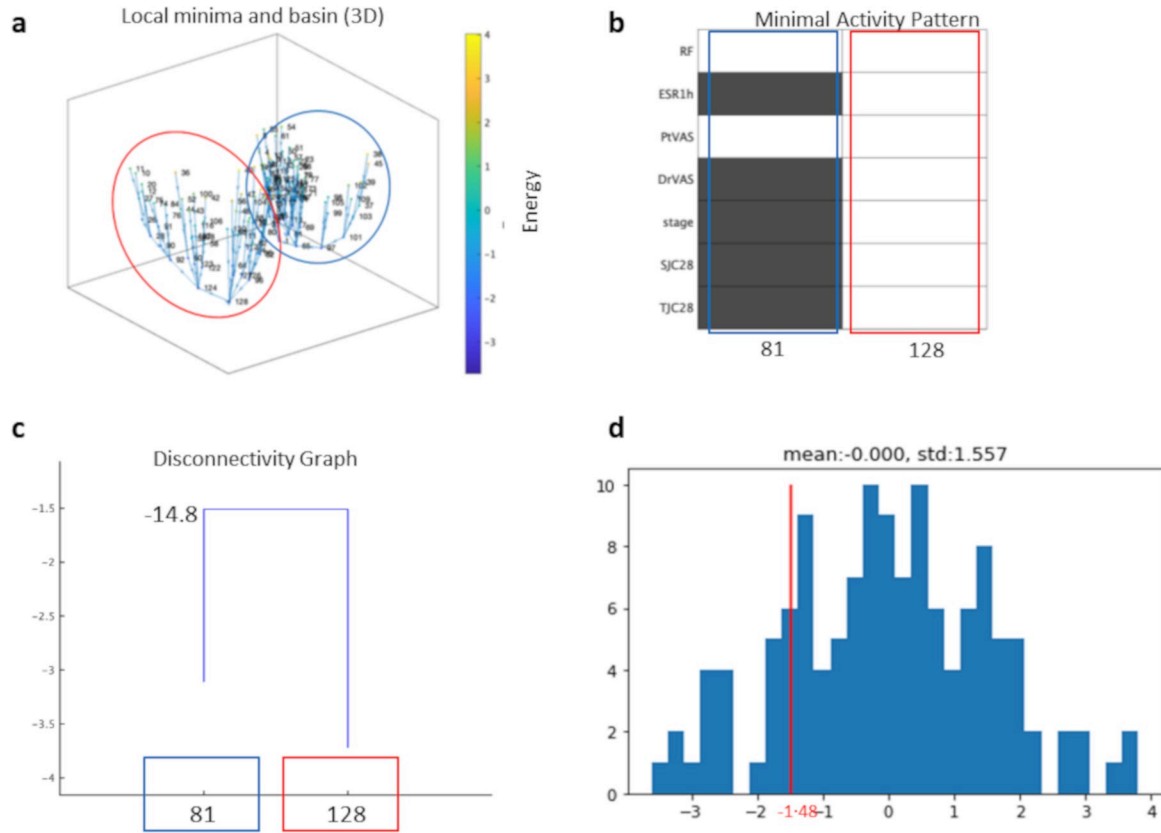

**Fig 4. Results of the energy landscape analysis.**

Fig 4D shows the distributions of state numbers and energies. The horizontal axis is the energy value, and the vertical axis is the number of states. The line in red indicates the energy threshold (-1.48) at which the pattern can move between states. The threshold value was approximately -1.48. Approximately 85% (109/128) of the state numbers had energy values that could be transferred between patterns.

The state numbers were aggregated into four quadrants at the energy threshold, and the number of patients per quadrant was calculated (Fig 5).

Fig 5A divides energy and pattern into four quadrants by threshold. The quadrant in which the energy is lower than the threshold and the activity pattern is "good stability" is G-L; the quadrant in which the energy is higher than the threshold and the activity pattern is "good stability" is G-H; the quadrant in which the energy is higher than the threshold and the activity pattern is "poor stability" is P-H; and the quadrant in which the energy is lower than the threshold and the activity pattern is "poor stability" is P-L.

Fig 5B shows the number of patients in each quadrant, with a marked increase in G-L and a decrease in P-L until approximately two points (i.e., 6 months), followed by a gradual increase or decrease. P-L decreased until approximately two points (i.e., 6 months), with no significant change thereafter. G-H decreased significantly throughout the entire period. G-H did not increase or decrease significantly across the entire treatment period.

Fig 5C shows the HAQ scores in each quadrant: G-L patients met the remission criteria for almost the entire period; the number of P-L patients who met the remission criteria decreased over time; the number of G-H and P-H patients who met the remission criteria changed

**Table 2. Factor breakdown per state number (High activity: Nonremission = 1; low activity: Remission = -1), energy per state number, next transition state number, and minimal state number.**

| State number | RF | ESR 1 h | Pt VAS | Dr VAS | STA GE | SJC 28 | TJC 28 | Energy | Next transition state number | Minimal state number |
|---|---|---|---|---|---|---|---|---|---|---|
| 1 | -1 | -1 | -1 | -1 | -1 | -1 | -1 | -2.7629 | 65 | 81 |
| 2 | -1 | -1 | -1 | -1 | -1 | -1 | 1 | 0.098937 | 1 | 81 |
| 3 | -1 | -1 | -1 | -1 | -1 | 1 | -1 | -0.19946 | 1 | 81 |
| 4 | -1 | -1 | -1 | -1 | -1 | 1 | 1 | 1.8936 | 3 | 81 |
| 5 | -1 | -1 | -1 | -1 | 1 | -1 | -1 | -0.79545 | 1 | 81 |
| 6 | -1 | -1 | -1 | -1 | 1 | -1 | 1 | 2.0551 | 5 | 81 |
| 7 | -1 | -1 | -1 | -1 | 1 | 1 | -1 | 0.94894 | 5 | 81 |
| 8 | -1 | -1 | -1 | -1 | 1 | 1 | 1 | 3.0307 | 7 | 81 |
| 9 | -1 | -1 | -1 | 1 | -1 | -1 | -1 | 0.84369 | 1 | 81 |
| 10 | -1 | -1 | -1 | 1 | -1 | -1 | 1 | 1.4102 | 26 | 128 |
| 11 | -1 | -1 | -1 | 1 | -1 | 1 | -1 | 1.1752 | 3 | 81 |
| 12 | -1 | -1 | -1 | 1 | -1 | 1 | 1 | 0.97285 | 28 | 128 |
| 13 | -1 | -1 | -1 | 1 | 1 | -1 | -1 | 2.2545 | 5 | 81 |
| 14 | -1 | -1 | -1 | 1 | 1 | -1 | 1 | 2.8096 | 30 | 128 |
| 15 | -1 | -1 | -1 | 1 | 1 | 1 | -1 | 1.7669 | 31 | 128 |
| 16 | -1 | -1 | -1 | 1 | 1 | 1 | 1 | 1.5532 | 32 | 128 |
| 17 | -1 | -1 | 1 | -1 | -1 | -1 | -1 | -2.8443 | 81 | 81 |
| 18 | -1 | -1 | 1 | -1 | -1 | -1 | 1 | -0.8946 | 17 | 81 |
| 19 | -1 | -1 | 1 | -1 | -1 | 1 | -1 | -0.39429 | 17 | 81 |
| 20 | -1 | -1 | 1 | -1 | -1 | 1 | 1 | 0.78658 | 28 | 128 |
| 21 | -1 | -1 | 1 | -1 | 1 | -1 | -1 | -1.4456 | 17 | 81 |
| 22 | -1 | -1 | 1 | -1 | 1 | -1 | 1 | 0.49277 | 21 | 81 |
| 23 | -1 | -1 | 1 | -1 | 1 | 1 | -1 | 0.18531 | 21 | 81 |
| 24 | -1 | -1 | 1 | -1 | 1 | 1 | 1 | 1.3548 | 32 | 128 |
| 25 | -1 | -1 | 1 | 1 | -1 | -1 | -1 | -0.37108 | 17 | 81 |
| 26 | -1 | -1 | 1 | 1 | -1 | -1 | 1 | -0.71684 | 90 | 128 |
| 27 | -1 | -1 | 1 | 1 | -1 | 1 | -1 | -0.15312 | 28 | 128 |
| 28 | -1 | -1 | 1 | 1 | -1 | 1 | 1 | -1.2677 | 92 | 128 |
| 29 | -1 | -1 | 1 | 1 | 1 | -1 | -1 | 0.47091 | 21 | 81 |
| 30 | -1 | -1 | 1 | 1 | 1 | -1 | 1 | 0.11382 | 94 | 128 |
| 31 | -1 | -1 | 1 | 1 | 1 | 1 | -1 | -0.13022 | 95 | 128 |
| 32 | -1 | -1 | 1 | 1 | 1 | 1 | 1 | -1.2561 | 96 | 128 |
| 33 | -1 | 1 | -1 | -1 | -1 | -1 | -1 | -1.421 | 1 | 81 |
| 34 | -1 | 1 | -1 | -1 | -1 | -1 | 1 | 1.484 | 33 | 81 |
| 35 | -1 | 1 | -1 | -1 | -1 | 1 | -1 | 0.59228 | 33 | 81 |
| 36 | -1 | 1 | -1 | -1 | -1 | 1 | 1 | 2.7285 | 35 | 81 |
| 37 | -1 | 1 | -1 | -1 | 1 | -1 | -1 | 0.49006 | 101 | 81 |
| 38 | -1 | 1 | -1 | -1 | 1 | -1 | 1 | 3.3837 | 37 | 81 |
| 39 | -1 | 1 | -1 | -1 | 1 | 1 | -1 | 1.6842 | 103 | 81 |
| 40 | -1 | 1 | -1 | -1 | 1 | 1 | 1 | 3.8091 | 39 | 81 |
| 41 | -1 | 1 | -1 | 1 | -1 | -1 | -1 | 1.6325 | 33 | 81 |
| 42 | -1 | 1 | -1 | 1 | -1 | -1 | 1 | 2.2421 | 58 | 128 |
| 43 | -1 | 1 | -1 | 1 | -1 | 1 | -1 | 1.4138 | 59 | 128 |
| 44 | -1 | 1 | -1 | 1 | -1 | 1 | 1 | 1.2546 | 60 | 128 |
| 45 | -1 | 1 | -1 | 1 | 1 | -1 | -1 | 2.9869 | 37 | 81 |
| 46 | -1 | 1 | -1 | 1 | 1 | -1 | 1 | 3.5851 | 62 | 128 |

*(Continued)*

**Table 2.** (Continued)

| State number | RF | ESR 1 h | Pt VAS | Dr VAS | STA GE | SJC 28 | TJC 28 | Energy | Next transition state number | Minimal state number |
|---|---|---|---|---|---|---|---|---|---|---|
| 47 | -1 | 1 | -1 | 1 | 1 | 1 | -1 | 1.949 | 63 | 128 |
| 48 | -1 | 1 | -1 | 1 | 1 | 1 | 1 | 1.7785 | 64 | 128 |
| 49 | -1 | 1 | 1 | -1 | -1 | -1 | -1 | -1.8425 | 113 | 81 |
| 50 | -1 | 1 | 1 | -1 | -1 | -1 | 1 | 0.15031 | 49 | 81 |
| 51 | -1 | 1 | 1 | -1 | -1 | 1 | -1 | 0.057273 | 49 | 81 |
| 52 | -1 | 1 | 1 | -1 | -1 | 1 | 1 | 1.2813 | 60 | 128 |
| 53 | -1 | 1 | 1 | -1 | 1 | -1 | -1 | -0.50023 | 117 | 81 |
| 54 | -1 | 1 | 1 | -1 | 1 | -1 | 1 | 1.4812 | 118 | 81 |
| 55 | -1 | 1 | 1 | -1 | 1 | 1 | -1 | 0.58042 | 119 | 81 |
| 56 | -1 | 1 | 1 | -1 | 1 | 1 | 1 | 1.7931 | 64 | 128 |
| 57 | -1 | 1 | 1 | 1 | -1 | -1 | -1 | 0.077572 | 49 | 81 |
| 58 | -1 | 1 | 1 | 1 | -1 | -1 | 1 | -0.22506 | 122 | 128 |
| 59 | -1 | 1 | 1 | 1 | -1 | 1 | -1 | -0.25468 | 123 | 128 |
| 60 | -1 | 1 | 1 | 1 | -1 | 1 | 1 | -1.3261 | 124 | 128 |
| 61 | -1 | 1 | 1 | 1 | 1 | -1 | -1 | 0.86311 | 125 | 81 |
| 62 | -1 | 1 | 1 | 1 | 1 | -1 | 1 | 0.54914 | 126 | 128 |
| 63 | -1 | 1 | 1 | 1 | 1 | 1 | -1 | -0.28824 | 127 | 128 |
| 64 | -1 | 1 | 1 | 1 | 1 | 1 | 1 | -1.371 | 128 | 128 |
| 65 | 1 | -1 | -1 | -1 | -1 | -1 | -1 | -3.1347 | 81 | 81 |
| 66 | 1 | -1 | -1 | -1 | -1 | -1 | 1 | -0.34836 | 65 | 81 |
| 67 | 1 | -1 | -1 | -1 | -1 | 1 | -1 | -0.43854 | 65 | 81 |
| 68 | 1 | -1 | -1 | -1 | -1 | 1 | 1 | 1.579 | 67 | 81 |
| 69 | 1 | -1 | -1 | -1 | 1 | -1 | -1 | -1.8916 | 65 | 81 |
| 70 | 1 | -1 | -1 | -1 | 1 | -1 | 1 | 0.88338 | 69 | 81 |
| 71 | 1 | -1 | -1 | -1 | 1 | 1 | -1 | -0.014557 | 69 | 81 |
| 72 | 1 | -1 | -1 | -1 | 1 | 1 | 1 | 1.9917 | 71 | 81 |
| 73 | 1 | -1 | -1 | 1 | -1 | -1 | -1 | 0.15463 | 65 | 81 |
| 74 | 1 | -1 | -1 | 1 | -1 | -1 | 1 | 0.64557 | 90 | 128 |
| 75 | 1 | -1 | -1 | 1 | -1 | 1 | -1 | 0.61881 | 91 | 128 |
| 76 | 1 | -1 | -1 | 1 | -1 | 1 | 1 | 0.34096 | 92 | 128 |
| 77 | 1 | -1 | -1 | 1 | 1 | -1 | -1 | 0.841 | 69 | 81 |
| 78 | 1 | -1 | -1 | 1 | 1 | -1 | 1 | 1.3206 | 94 | 128 |
| 79 | 1 | -1 | -1 | 1 | 1 | 1 | -1 | 0.48609 | 95 | 128 |
| 80 | 1 | -1 | -1 | 1 | 1 | 1 | 1 | 0.19691 | 96 | 128 |
| 81 | 1 | -1 | 1 | -1 | -1 | -1 | -1 | -3.1748 | 81 | 81 |
| 82 | 1 | -1 | 1 | -1 | -1 | -1 | 1 | -1.3007 | 81 | 81 |
| 83 | 1 | -1 | 1 | -1 | -1 | 1 | -1 | -0.59216 | 81 | 81 |
| 84 | 1 | -1 | 1 | -1 | -1 | 1 | 1 | 0.51318 | 92 | 128 |
| 85 | 1 | -1 | 1 | -1 | 1 | -1 | -1 | -2.5005 | 81 | 81 |
| 86 | 1 | -1 | 1 | -1 | 1 | -1 | 1 | -0.63775 | 85 | 81 |
| 87 | 1 | -1 | 1 | -1 | 1 | 1 | -1 | -0.73698 | 85 | 81 |
| 88 | 1 | -1 | 1 | -1 | 1 | 1 | 1 | 0.35703 | 96 | 128 |
| 89 | 1 | -1 | 1 | 1 | -1 | -1 | -1 | -1.0189 | 81 | 81 |
| 90 | 1 | -1 | 1 | 1 | -1 | -1 | 1 | -1.4402 | 122 | 128 |
| 91 | 1 | -1 | 1 | 1 | -1 | 1 | -1 | -0.66827 | 92 | 128 |
| 92 | 1 | -1 | 1 | 1 | -1 | 1 | 1 | -1.8583 | 124 | 128 |
| 93 | 1 | -1 | 1 | 1 | 1 | -1 | -1 | -0.90136 | 85 | 81 |

(*Continued*)

**Table 2.** (Continued)

| State number | RF | ESR 1 h | Pt VAS | Dr VAS | STA GE | SJC 28 | TJC 28 | Energy | Next transition state number | Minimal state number |
|---|---|---|---|---|---|---|---|---|---|---|
| 94 | 1 | -1 | 1 | 1 | 1 | -1 | 1 | -1.334 | 96 | 128 |
| 95 | 1 | -1 | 1 | 1 | 1 | 1 | -1 | -1.3698 | 96 | 128 |
| 96 | 1 | -1 | 1 | 1 | 1 | 1 | 1 | -2.5712 | 128 | 128 |
| 97 | 1 | 1 | -1 | -1 | -1 | -1 | -1 | -2.7062 | 65 | 81 |
| 98 | 1 | 1 | -1 | -1 | -1 | -1 | 1 | 0.12328 | 97 | 81 |
| 99 | 1 | 1 | -1 | -1 | -1 | 1 | -1 | -0.56024 | 97 | 81 |
| 100 | 1 | 1 | -1 | -1 | -1 | 1 | 1 | 1.5004 | 99 | 81 |
| 101 | 1 | 1 | -1 | -1 | 1 | -1 | -1 | -1.5196 | 97 | 81 |
| 102 | 1 | 1 | -1 | -1 | 1 | -1 | 1 | 1.2986 | 101 | 81 |
| 103 | 1 | 1 | -1 | -1 | 1 | 1 | -1 | -0.19272 | 101 | 81 |
| 104 | 1 | 1 | -1 | -1 | 1 | 1 | 1 | 1.8566 | 112 | 128 |
| 105 | 1 | 1 | -1 | 1 | -1 | -1 | -1 | 0.030022 | 97 | 81 |
| 106 | 1 | 1 | -1 | 1 | -1 | -1 | 1 | 0.56409 | 122 | 128 |
| 107 | 1 | 1 | -1 | 1 | -1 | 1 | -1 | -0.056019 | 123 | 128 |
| 108 | 1 | 1 | -1 | 1 | -1 | 1 | 1 | -0.29074 | 124 | 128 |
| 109 | 1 | 1 | -1 | 1 | 1 | -1 | -1 | 0.65994 | 101 | 81 |
| 110 | 1 | 1 | -1 | 1 | 1 | -1 | 1 | 1.1827 | 126 | 128 |
| 111 | 1 | 1 | -1 | 1 | 1 | 1 | -1 | -0.2452 | 127 | 128 |
| 112 | 1 | 1 | -1 | 1 | 1 | 1 | 1 | -0.49126 | 128 | 128 |
| 113 | 1 | 1 | 1 | -1 | -1 | -1 | -1 | -3.0865 | 81 | 81 |
| 114 | 1 | 1 | 1 | -1 | -1 | -1 | 1 | -1.1692 | 113 | 81 |
| 115 | 1 | 1 | 1 | -1 | -1 | 1 | -1 | -1.054 | 113 | 81 |
| 116 | 1 | 1 | 1 | -1 | -1 | 1 | 1 | 0.094419 | 124 | 128 |
| 117 | 1 | 1 | 1 | -1 | 1 | -1 | -1 | -2.4687 | 113 | 81 |
| 118 | 1 | 1 | 1 | -1 | 1 | -1 | 1 | -0.56275 | 117 | 81 |
| 119 | 1 | 1 | 1 | -1 | 1 | 1 | -1 | -1.2553 | 117 | 81 |
| 120 | 1 | 1 | 1 | -1 | 1 | 1 | 1 | -0.11819 | 128 | 128 |
| 121 | 1 | 1 | 1 | 1 | -1 | -1 | -1 | -1.4837 | 113 | 81 |
| 122 | 1 | 1 | 1 | 1 | -1 | -1 | 1 | -1.8619 | 124 | 128 |
| 123 | 1 | 1 | 1 | 1 | -1 | 1 | -1 | -1.6833 | 124 | 128 |
| 124 | 1 | 1 | 1 | 1 | -1 | 1 | 1 | -2.8302 | 128 | 128 |
| 125 | 1 | 1 | 1 | 1 | 1 | -1 | -1 | -1.4226 | 117 | 81 |
| 126 | 1 | 1 | 1 | 1 | 1 | -1 | 1 | -1.8121 | 128 | 128 |
| 127 | 1 | 1 | 1 | 1 | 1 | 1 | -1 | -2.4413 | 128 | 128 |
| 128 | 1 | 1 | 1 | 1 | 1 | 1 | 1 | -3.5995 | 128 | 128 |

minimally over time; and the number of P-H patients who met the criteria did not change over time.

In terms of overall treatment, the number of patients who responded to drug therapy and who demonstrated an improved disease status increased markedly until approximately 6 months after the start of treatment, after which the rate of increase gradually slowed. The rate of change plateaued after 1 year.

## Results of time-series clustering

Individual patient state transitions are highly variable, making it difficult to identify patterns across the population (S1 Appendix). Time-series clustering was performed based on activity

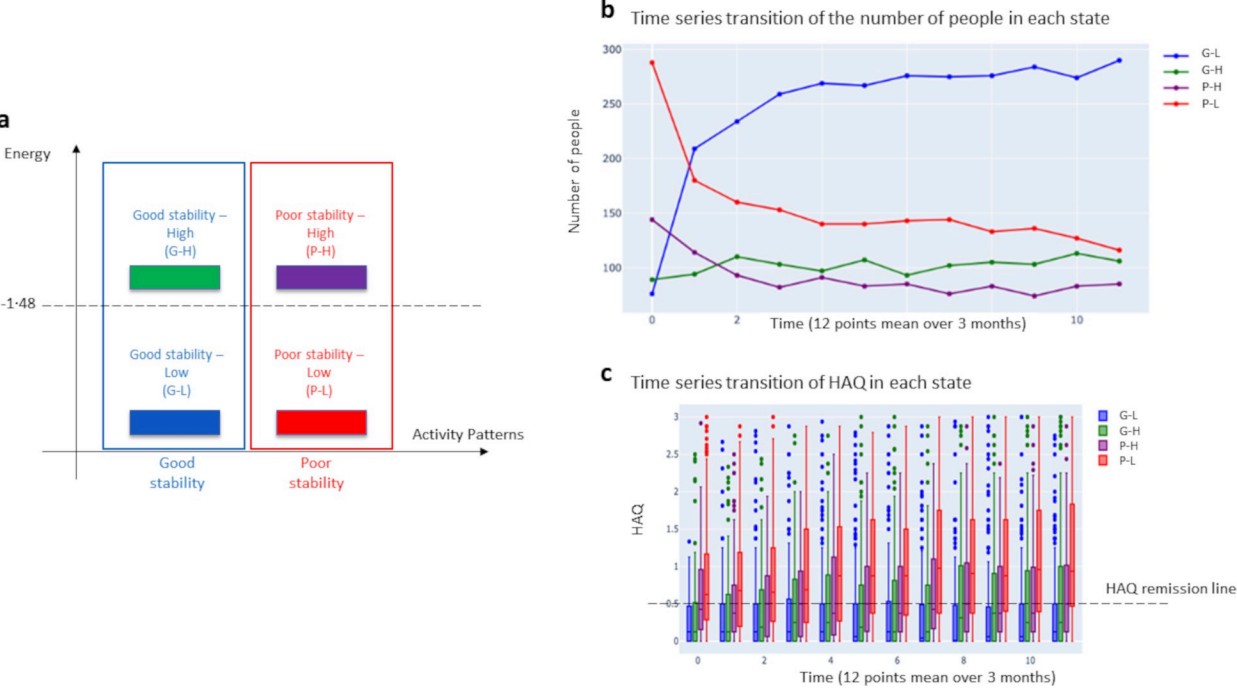

**Fig 5. Remission trends by number of people and HAQ score in the four quadrants.**

patterns, energy, and HAQ scores, and patient profiling was performed for each cluster. The number of clusters was set to "3" based on comprehensive analysis using the elbow method and silhouette technique [48, 49] (Fig 6 and Table 3).

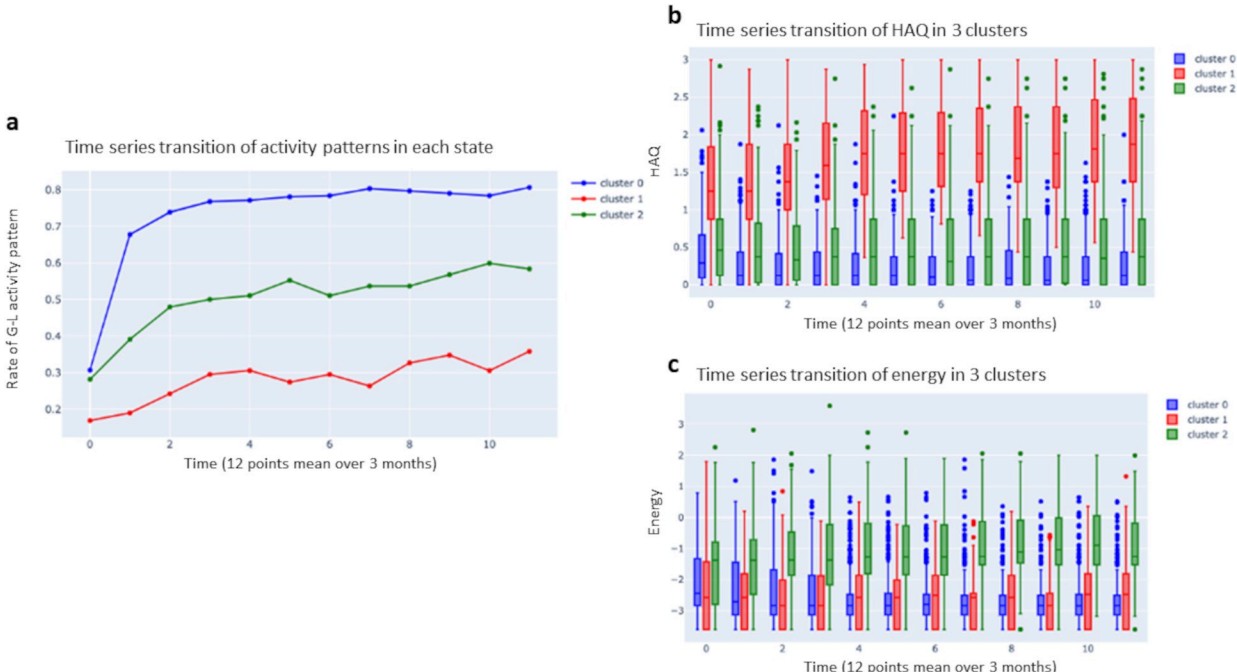

**Fig 6. Results of time-series clustering.** (A) Percent changes in the numbers of patients in the following three clusters: cluster 0, "toward good stability"; cluster 1, "toward poor stability"; and cluster 2, "unstable." (B) HAQ scores. c Energy values.

**Table 3. Detailed characteristics of patients in the three clusters, based on data obtained at initial examination.**

| | | Cluster 0: toward good stability | Cluster 1: toward poor stability | Cluster 2: unstable |
|---|---|---|---|---|
| Sex | | | | |
| | Male, n (%) | 71 (22.9%) | 14 (14.7%) | 38 (19.8%) |
| | Female, n (%) | 239 (77.1%) | 81 (85.3%) | 154 (80.2%) |
| Age, laboratory values at initial examination, etc. | | | | |
| | Age at initial examination (years) | 58.9 ± 12.9 | 69.2 ± 9.2 | 60.6 ± 13.4 |
| | Duration of illness (years) | 2.5 (0.1–47.0) | 17.3 (0.1–62.4) | 5.3 (0.1–53.7) |
| | Age at disease onset (years) | 52.1 ± 14.0 | 51.5 ± 16.7 | 51.6 ± 15.7 |
| | HAQ | 0.6 ± 0.6 | 1.4 ± 0.8 | 0.6 ± 0.7 |
| | mHAQ | 0.4 ± 0.5 | 1.1 ± 0.7 | 0.5 ± 0.5 |
| | ACPA (U/mL) | 73.7 (0.5–1890.0) | 107 (0.6–1140.0) | 54.8 (0.5–1960.0) |
| | RF (IU/mL) | 59.2 (6.0–2173.8) | 119.1 (8.0–1466.8) | 27.1 (6.0–889.9) |
| | CRP (mg/dL) | 0.4 (0.0–23.4) | 0.9 (0.0–11.8) | 0.3 (0.0–12.2) |
| | ESR (mm/h) | 23 (0.0–111.0) | 35 (2.0–124.0) | 20 (1.0–114.0) |
| | CDAI | 11.7 (0.1–53.5) | 13.8 (0.5–52.8) | 11.4 (0.0–66.0) |
| | SDAI | 13.4 (0.2–71.8) | 15.5 (1.0–60.4) | 13 (0.2–70.1) |
| | DAS28-ESR | 4.0 ± 1.5 | 4.5 ± 1.3 | 4.0 ± 1.5 |
| | DAS28-CRP | 3.5 ± 1.4 | 3.9 ± 1.2 | 3.4 ± 1.4 |
| Stage, Class | | | | |
| | Stage 1, 2, 3, 4 | 45.80%, 29.70%, 8.40%, 16.10% | 10.50%, 10.50%, 21.10%, 57.90% | 25.00%, 24.50%, 21.90%, 28.60% |
| | Class 1, 2, 3, 4 | 23.50%, 67.10%, 9.00%, 0.30%, | 3.20%, 51.60%, 42.10%, 3.20% | 25.50%, 62.50%, 10.90%, 1.00% |

Cluster 0 (toward good stability) (blue) was defined as patients with low energy, 80% of whom eventually assumed a good stability pattern and became stable. Cluster 1 (toward poor stability) (red) was defined as patients with low energy, almost all of whom were characterized by the "poor stability" pattern and had high HAQ scores. Cluster 2 (unstable) (green) consisted of patients with relatively high energy, almost all of whom demonstrated a "poor stability" pattern in the early stage of treatment. For the patients in Cluster 2 (unstable)", the treatment response was not yet determined, and the majority of them met the remission criterion based on the HAQ score, although a high proportion of these patients exhibited a "poor stability" pattern in the early stage of treatment. Although the age at disease onset did not differ significantly between clusters, the age at initial diagnosis, duration of disease, and laboratory tests at initial diagnosis did differ, with the "poor stability" cluster (Cluster 1) showing a greater tendency for age and all laboratory values at initial examination than the other clusters.

The state transitions of individuals from each cluster are shown in the S2 Appendix.

## Discussion

### Key results

The whole treatment course of patients in the KURAMA cohort was analyzed in a time-dependent manner.

Energy landscape analysis revealed that state transitions were divided into two patterns: "good stability leading to remission" and "poor stability below the energy threshold without functional remission (i.e., treatment dead-end)." The energy threshold cutoff for switching

between patterns was approximately -1.48. The states were aggregated into four quadrants based on this energy threshold, and the number of patients in each quadrant was calculated. The number of patients who demonstrated status improvement increased markedly until approximately 6 months after the start of treatment, after which the rate of increase gradually slowed. The rate of change plateaued after 1 year.

Patient profiling by time-series clustering showed that patients were classified into three clusters: "toward good stability," "toward poor stability," and "unstable." Patients in the "unstable" cluster are considered to have clinical courses that are difficult to predict in RA practice; therefore, these patients should be treated with more care. Age at initial diagnosis, duration of disease, and laboratory values at initial diagnosis tended to differ by cluster, with the "toward poor stability" cluster exhibiting a higher age and higher laboratory values at initial examination than the other clusters. Compared to the "toward good stability" cluster, the "unstable" cluster had a higher age at initial examination and a longer duration of illness, but the "unstable" cluster tended to have better laboratory values at initial examination. The difference between these two clusters was marginal; however, there was a possibility that the initiation of effective treatment was delayed in the "unstable" cluster due to the relatively favorable initial conditions at the time of the first visit. Further research is warranted; however, the results of this study indicate the importance of early disease detection and treatment initiation, especially in "unstable" cases.

Energy landscape analysis was performed to visualize the state transitions of the patient population as a whole. However, this approach alone did not allow for the visualization of individual patient state transitions. We utilized time-series clustering to clarify the specific state transitions occurring within the given population. These methodologies complemented each other, revealing insights into both the collective state transitions of the population and the characteristic attributes of patients within that population. Traditionally, treatment decisions have been made based on past treatment histories and the latest evaluation of disease activity. Energy landscape analysis and time-series clustering analysis could yield insights regarding how a patient's state might transition in the future and the optimal timing for effective treatment.

## Limitations

This was a single-center observational study at Kyoto University Hospital that used registry data from 2011 to 2018. This study also did not evaluate the effectiveness of state transition visualization in daily RA drug therapy.

## Interpretation

Currently, RA treatment practices are based on practice guidelines, and treatment approaches are determined based on evaluations of the efficacies of previously administered agents. Specific treatment strategies for individual patients are determined based on data obtained from comprehensive disease activity assessments and on the treatment history from the initial diagnosis to the present, and future state transitions rely on speculation based on physicians' clinical experience. In our study, patient response to drug therapy and improved disease status significantly increased for approximately 6 months after the start of treatment and then plateaued after 1 year. Furthermore, our findings demonstrated that the patient population could be divided into three clusters: "toward good stability," "toward poor stability," and "unstable." We quantitatively clarified the timing and duration of significant treatment effects through time-dependent analysis of the state of patients with RA, which is ever-changing due to drug therapy. In addition to the conventional assessment of overall disease activity, the evaluation of energy (i.e., state multistability) enables us to understand the relevance of the current state

in the overall state transition related to RA drug treatment and to predict future state transitions. In the realm of artificial intelligence (AI) applications in healthcare, there is a fascinating parallel with the domain of chess, shogi, and go AI, where professionals occasionally find themselves defeated by seemingly unorthodox moves beyond human comprehension [52]. Similarly, in the context of real-world practice visualization derived from our methodologies, the recommended treatment may significantly deviate from established clinical guidelines. Despite low disease activity, AI could suggest early and intensive administration of biologic agents. In such cases, it becomes imperative to assume that the proposed therapeutic approach of AI holds merit and to therefore perform rigorous validation through randomized controlled trials (RCTs). Our research suggests the emergence of an innovative approach to the future landscape of medical research and practice, tentatively termed "AI-based RCTs," to systematically investigate and corroborate the efficacy of AI-suggested clinical interventions. In the context of our study, for patients who are likely to experience treatment dead-end, it may be necessary to select a more effective treatment, such as a biologic, at an earlier stage. For patients whose condition improves, it may be possible to offer drug discontinuation or dose reduction. Clinical research involving "unstable" patients may allow for more effective clinical evaluation with smaller numbers of patients. Furthermore, state transition visualization may be a useful tool for facilitating communication between physicians and patients. Ultimately, by selecting treatments according to a patient's cluster and their state transitions, it may be possible to adopt a personalized medicine approach targeting each patient's state. This study suggested the possibility of optimizing the treatment plan based on the whole treatment course.

## Generalizability

This was a single-center observational study. Kyoto University Hospital is a tertiary hospital and may accommodate patients with more severe illness than general hospitals and clinics; therefore, a multicenter study is needed. In addition, since this study was based on data collected in 2018, new drugs such as JAK inhibitors were likely to be used less frequently. Therefore, data from 2019 and beyond should be examined in future studies. Analyzing more recent data will enable the visualization of state transitions of RA drug therapy that more accurately reflect the reality of daily practice.

We believe that our proposed approach is useful as a visualization method for multidimensional time-series data in medicine and can be applied to diseases other than RA. In general, it is difficult to intuitively interpret multidimensional time-series data in medicine. We have made it possible to readily interpret state transitions in the drug treatment of patients with RA by consolidating the seven factors used in comprehensive disease activity assessment in daily practice into two dimensions, namely, state number and energy, and clustering along the time course. In many diseases, even the best treatment does not lead to a cure or remission; therefore, conservative treatment is used to prevent deterioration and maintain a good state. For multifactorial diseases such as diabetes and hypertension, target treatment values such as HbA1C and blood pressure levels are set by guidelines, and treatment strategies are chosen based on concepts equivalent to T2T in RA [53, 54]. Various factors, such as diet, lack of exercise, and stress, affect the condition of patients with multifactorial diseases. Nonetheless, advances in digital health technology have led to increased adoption of personal health records and health-related apps; therefore, the collection of frequently measured time-series sensor data from wearable devices is becoming increasingly feasible [55–57]. We believe that our method may provide useful information for optimizing treatment plans to achieve behavioral change and personalized medicine by reducing the dimensionality of multiple factors and visualizing state transitions.

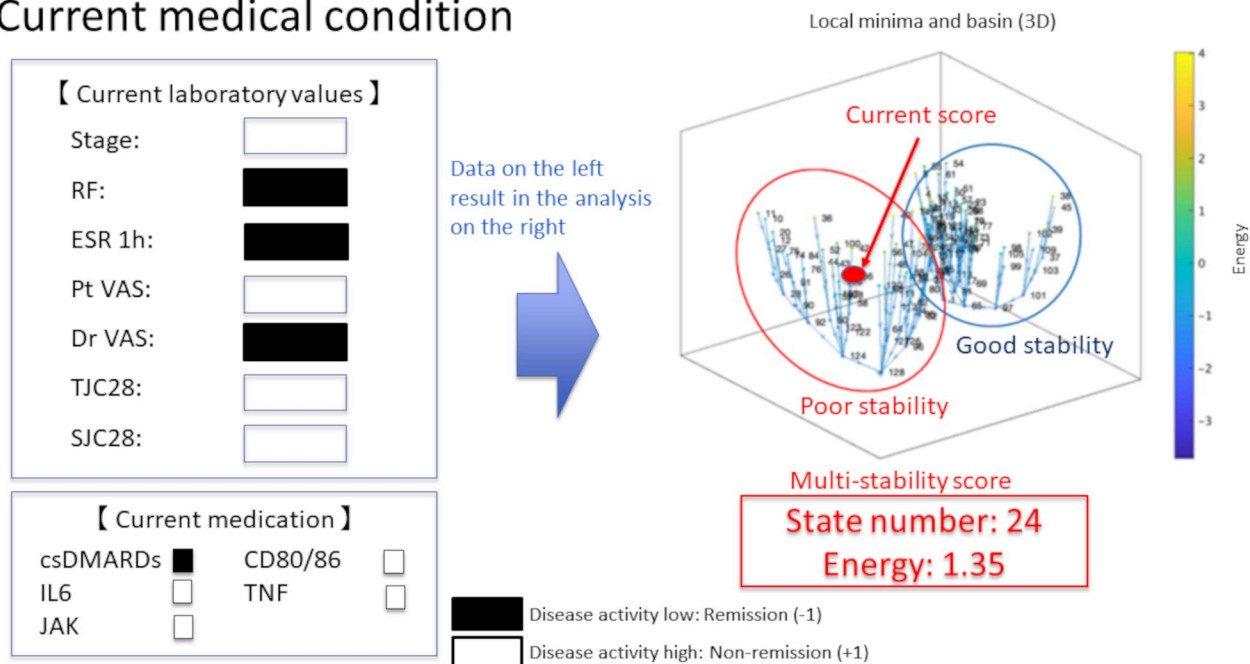

**Fig 7. Examples of future real-world clinical implementations.** After the inspection is completed, the seven factors are entered into a tablet by the physician, and the state number, energy, and coordinate of the multistability score in the state transition are displayed. In this example, the energy is higher than the threshold value and is in a transitional state; therefore, the prospects for remission are considered adequate. However, the patient had a poor-stability cluster and is currently using csDMARDs. A change to a more effective drug may be considered.

Considering the application of this research to personalized medicine in daily practice, it is necessary to develop an application that can calculate energy based on these seven factors and display its position in the state transition pattern in real time (Fig 7).

In addition, standardization of medical practice is necessary for high-quality multidimensional time-series data collection. Conventional medical practice involves interviewing each patient and performing necessary tests on a case-by-case basis. In the KURAMA cohort, all physicians used a common medical questionnaire to record patient states over time under certain standards and quality controls. It is necessary to standardize data and improve the medical system to enable the collection of high-quality time-series data that are useful for both medical care and analysis while balancing routine medical care and data analysis and eliminating unnecessary burdens as much as possible [58, 59].

## Conclusions

This study suggested that evaluating state multistability and determining the patient's state in daily practice may enable treatment plan optimization over the entire course of treatment. We believe that this study will contribute to the development of personalized medicine utilizing real-world data.

## Supporting information

**S1 Checklist. STROBE statement—checklist of items that should be included in reports of observational studies.**
(DOC)

**S1 Table. List of the state transition probabilities of all 597 patients.**
(DOCX)

**S1 Appendix. The state transitions of 10 randomly selected individuals.** The vertical axis indicates high and low energy. For the horizontal axis, the central vertical line divides the groups according to whether they are likely to transition to good stability or to persist in poor stability, according to the energy landscape analysis, and the solid red vertical line indicates the energy threshold (i.e., -1.48).
(DOCX)

**S2 Appendix. The state transitions of 10 randomly selected individuals from each cluster.** Transitions in the good stability (A), poor stability (B), and unstable (C) clusters. Individuals in cluster 1 (poor stability) tend to have low energy and to remain in the poor stability quadrant. On the other hand, those in cluster 2 (unstable) generally have higher energy and tend to move between the good stability and poor stability quadrants.
(DOCX)

## Acknowledgments

The authors would like to express their deepest gratitude to Dr. Satoshi Teramukai of Kyoto Prefectural University of Medicine, the staff of Wakayama Medical University, and Dr. Shunsuke Baba of Osaka Dental University for their helpful advice in conducting this study. We would like to extend our heartfelt gratitude to Haruo Horii, Naohiro Ito, and Masatoshi Fujii for their generous support for the KURAMA cohort.

## Author Contributions

**Conceptualization:** Keiichi Yamamoto, Masahiko Sakaguchi, Akira Onishi, Shinichiro Yokoyama, Yusuke Matsui, Wataru Yamamoto, Motomu Hashimoto, Shuichi Matsuda, Akio Morinobu.

**Data curation:** Keiichi Yamamoto, Shinichiro Yokoyama, Yusuke Matsui, Wataru Yamamoto, Hideo Onizawa, Takayuki Fujii, Koichi Murata, Masao Tanaka.

**Formal analysis:** Keiichi Yamamoto, Masahiko Sakaguchi, Akira Onishi, Shinichiro Yokoyama, Yusuke Matsui.

**Funding acquisition:** Keiichi Yamamoto.

**Investigation:** Keiichi Yamamoto, Akira Onishi, Wataru Yamamoto, Hideo Onizawa, Takayuki Fujii, Koichi Murata, Masao Tanaka, Motomu Hashimoto.

**Methodology:** Keiichi Yamamoto, Masahiko Sakaguchi, Akira Onishi, Shinichiro Yokoyama, Yusuke Matsui, Wataru Yamamoto, Motomu Hashimoto.

**Project administration:** Keiichi Yamamoto.

**Resources:** Keiichi Yamamoto.

**Software:** Keiichi Yamamoto, Shinichiro Yokoyama, Yusuke Matsui.

**Supervision:** Keiichi Yamamoto, Motomu Hashimoto, Shuichi Matsuda, Akio Morinobu.

**Validation:** Keiichi Yamamoto, Masahiko Sakaguchi, Akira Onishi, Shinichiro Yokoyama, Yusuke Matsui, Wataru Yamamoto, Hideo Onizawa, Takayuki Fujii, Koichi Murata, Masao Tanaka, Motomu Hashimoto.

**Visualization:** Keiichi Yamamoto, Masahiko Sakaguchi, Shinichiro Yokoyama, Yusuke Matsui.

**Writing – original draft:** Keiichi Yamamoto, Masahiko Sakaguchi.

**Writing – review & editing:** Keiichi Yamamoto, Masahiko Sakaguchi, Akira Onishi, Shinichiro Yokoyama, Yusuke Matsui, Wataru Yamamoto, Hideo Onizawa, Takayuki Fujii, Koichi Murata, Masao Tanaka, Motomu Hashimoto, Shuichi Matsuda, Akio Morinobu.

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
