## [Decision Letter · Decision Letter 0]

20 Nov 2023

PONE-D-23-27887Energy Landscape Analysis and Time-series Clustering Characterizes Patient State Multi-stability Related to Rheumatoid Arthritis Drug Treatment: The KURAMA Cohort StudyPLOS ONE

Dear Dr. Yamamoto,

Thank you for submitting your manuscript to PLOS ONE. After careful consideration, we feel that it has merit but does not fully meet PLOS ONE’s publication criteria as it currently stands. Therefore, we invite you to submit a revised version of the manuscript that addresses the points raised during the review process. Specifically, both reviewers found some interests in this study, but also pointed out a number of concerns that require improvement or amendment. I request you to respond to all comments made by reviewers in the revised version. 

We look forward to receiving your revised manuscript.

Kind regards,

Masataka Kuwana, MD, PhD

Academic Editor

PLOS ONE

Journal Requirements:

"The Department of Advanced Medicine for Rheumatic Diseases, Kyoto University Graduate School of Medicine, is supported by Nagahama City, Shiga, Japan; Toyooka City, Hyogo, Japan; Asahi Kasei Pharma Corp.; and AYUMI Pharmaceutical Co. MH received research grants and/or speaker fees from Abbvie, Asahi Kasei, Astellas, Ayumi, Brystol Meyers, Chugai, EA Pharma, Eisai, Daiichi Sankyo, Eli Lilly, Novartis Pharma, and Tanabe Mitsubishi. M.T. received research grants and/or speaker fees from AbbVie GK, Asahi Kasei Pharma Corp., Astellas Pharma Inc., Chugai Pharmaceutical Co., Ltd., Daiichi Sankyo Co., Ltd., Eisai Co., Ltd., Eli Lilly Japan K.K., Janssen Pharmaceutical K.K., Kyowa Kirin Co., Ltd., Pfizer Inc., Taisho Pharmaceutical Co., Ltd., Tanabe Mitsubishi Pharma Corp., Teijin Pharma, Ltd., and UCB Japan Co., Ltd. K.M. received speaking and/or consulting fees from AbbVie GK, Eisai Co., Ltd., Pfizer Inc., Chugai Pharmaceutical Co., Ltd., Mitsubishi Tanabe Pharma Corp., Bristol-Myers Squibb, Daiichi Sankyo Co., Ltd., Janssen Pharmaceutical K.K., and Asahi Kasei Pharma Corp. The other authors declare that they have no conflicts of interest."

3. We notice that your supplementary [S1 Table] are included in the manuscript file. Please remove them and upload them with the file type 'Supporting Information'. Please ensure that each Supporting Information file has a legend listed in the manuscript after the references list.

Reviewers' comments:

Reviewer's Responses to Questions

**Comments to the Author**

1. Is the manuscript technically sound, and do the data support the conclusions?

Reviewer #1: Partly

Reviewer #2: Yes

2. Has the statistical analysis been performed appropriately and rigorously? 

Reviewer #1: I Don't Know

Reviewer #2: Yes

3. Have the authors made all data underlying the findings in their manuscript fully available?

Reviewer #1: No

Reviewer #2: Yes

4. Is the manuscript presented in an intelligible fashion and written in standard English?

Reviewer #1: Yes

Reviewer #2: Yes

5. Review Comments to the Author

Reviewer #1: The reviewer thinks this is an interesting analysis method that takes into account time variation.

Major comments:

1. The approach of combining two analysis methods is novel. The authors need to expound upon the reasoning behind merging these two methods. How do the two methods' shortcomings balance each other out when combined?

2. Using "energy" as a descriptor for the change in disease activity is intriguing. While the results seem to be clinically consistent with evaluating changes in disease activity, what fresh insights does this study offer by adopting such a perspective?

3. In the "Interpretation" section, the authors highlight the quantitative elucidation of significant treatment effects in terms of timing and duration. Please specify which particular result or findings you are referencing here.

4. Furthermore, in the "Interpretation" section, there is a suggestion that for certain patients who might be heading towards a treatment dead-end, a more potent treatment choice could be essential. Given that this study is anchored in a T2T-based approach, is it conceivable that such enhanced treatment decisions could be made?

Minor comments:

1. The units of measurement must be explicitly stated in the table for clarity.

2. On Page 5, Line 50, the statement "The importance of early disease detection and treatment initiation." stands incomplete. Kindly provide context or rephrase for completeness.

Reviewer #2: Dear authors,

The uniqueness of this manuscript was that the authors used energy landscape analysis and time-series clustering, which are methods different from common multivariate regression analysis, to examine the stability- and instability-based behavior of RA patients. This methodology and findings are interesting and the manuscript has a potential of interest, however, there are some concerns on this manuscript.

1． The major problem of this manuscript is the definition of the terms "Stability" and "Remission". The authors should better to make sure the definitions and keep those terms consistently throughout the manuscript.

(1) The term "remission" appears in this manuscript at several different situations and each implies a different meaning. For example, the word "remission" is used in "good stability leading to remission" and it used in each of the variables used in the energy landscape analysis. Additionally, the definition of "good stability" is "functional remission". From the above, it is necessary that the readers should guess the meaning of "remission" each time when it appears. I suggest that the authors consider improving on this terminology of “remission”.

(2) In the methods section, the authors described that, “Good stability” was defined as meeting functional remission criteria based on the Health Assessment Questionnaire (HAQ), and “poor stability” was defined as dying or falling below the energy threshold without functional remission. This complicated definition makes it very difficult to understand Figure 5. According to the definition, G-L seems to be a combination of good stability = functionally good state and Low-energy = stability (+), which is still understandable. However, the most difficult one is the P-H combination, which is a combination of poor stability = functionally poor with falling below the threshold of energy (stability +) and High-energy = stability (-). This was quite confusing for the readers. I think that the "stability" is essentially defined from the result of high or low energy by energy landscape analysis.

(3) The authors probably described Poor as Bad initially. I have found remnants of this in the text and in the Figure caption, Table. This made it very difficult to read, especially in the case of abbreviations such as B-L, as I could not keep up with their understanding. Please make sure to use the word "Poor" in a consistent manner.

(4) Please clarify 1) and 2) above and then rewrite Figure 1. This is the figure that confused me the most because the terms of “Good or Poor stability”, “remission” are mixed up in this figure. As a minor point, the word "Energy-igh" in the figure is probably a typo for -high, so please correct it.

2．In Figure 5, a 4x4 analysis is performed using the high and low scores derived from the energy landscape analysis and the good or poor stability defined by the HAQ. Is there any correlation or relationship between the total value of each activity variables used in the energy landscape analysis and HAQ?

6. PLOS authors have the option to publish the peer review history of their article (what does this mean?). If published, this will include your full peer review and any attached files.

Reviewer #1: No

Reviewer #2: **Yes: **Yasushi Kondo

---

## [Author Response · Author response to Decision Letter 0]

28 Feb 2024

Response to Reviewers: 

We thank the reviewers for their comments. Below are our point-by-point responses. For each response, please refer to the relevant page and line number(s) in the Revised Manuscript with Track Changes file.

Journal Requirements: 

1) Please ensure that your manuscript meets PLOS ONE's style requirements, including those for file naming. The PLOS ONE style templates can be found at https://journals.plos.org/plosone/s/file?id=wjVg/PLOSOne_formatting_sample_main_body.pdf and https://journals.plos.org/plosone/s/file?id=ba62/PLOSOne_formatting_sample_title_authors_affiliations.pdf

Response:

Thank you for the guidance. We have checked that our paper meets PLOS ONE’s style requirements.

2) Thank you for stating the following in the Competing Interests section: "The Department of Advanced Medicine for Rheumatic Diseases, Kyoto University Graduate School of Medicine, is supported by Nagahama City, Shiga, Japan; Toyooka City, Hyogo, Japan; Asahi Kasei Pharma Corp.; and AYUMI Pharmaceutical Co. MH received research grants and/or speaker fees from Abbvie, Asahi Kasei, Astellas, Ayumi, Brystol Meyers, Chugai, EA Pharma, Eisai, Daiichi Sankyo, Eli Lilly, Novartis Pharma, and Tanabe Mitsubishi. M.T. received research grants and/or speaker fees from AbbVie GK, Asahi Kasei Pharma Corp., Astellas Pharma Inc., Chugai Pharmaceutical Co., Ltd., Daiichi Sankyo Co., Ltd., Eisai Co., Ltd., Eli Lilly Japan K.K., Janssen Pharmaceutical K.K., Kyowa Kirin Co., Ltd., Pfizer Inc., Taisho Pharmaceutical Co., Ltd., Tanabe Mitsubishi Pharma Corp., Teijin Pharma, Ltd., and UCB Japan Co., Ltd. K.M. received speaking and/or consulting fees from AbbVie GK, Eisai Co., Ltd., Pfizer Inc., Chugai Pharmaceutical Co., Ltd., Mitsubishi Tanabe Pharma Corp., Bristol-Myers Squibb, Daiichi Sankyo Co., Ltd., Janssen Pharmaceutical K.K., and Asahi Kasei Pharma Corp. The other authors declare that they have no conflicts of interest.".

Response:

Thank you for the guidance. We included our updated Competing Interests statement in our cover letter.

3) We notice that your supplementary [S1 Table] are included in the manuscript file. Please remove them and upload them with the file type 'Supporting Information'. Please ensure that each Supporting Information file has a legend listed in the manuscript after the references list.

Response:

Thank you for the guidance. We have removed them and upload a supporting information file.

Reviewer #1

Major comments:

1) The approach of combining two analysis methods is novel. The authors need to expound upon the reasoning behind merging these two methods. How do the two methods' shortcomings balance each other out when combined?

Response:

Thank you for your valuable assessment. The energy landscape analysis is intended to visualize the state transitions of the patient population as a whole. However, this approach alone does not allow for the visualization of individual patient state transitions. Therefore, we utilized time-series clustering to clarify the specific state transitions occurring within the given population. We believe that these respective methodologies complement each other, revealing insights into both the collective state transitions of the population and the characteristic attributes of patients within that population. We have added these sentences to the “Key results” section (page 27, line 412 to 417).

2) Using "energy" as a descriptor for the change in disease activity is intriguing. While the results seem to be clinically consistent with evaluating changes in disease activity, what fresh insights does this study offer by adopting such a perspective?

Response:

Thank you for your valuable assessment. Traditionally, treatment decisions have been made based on past treatment histories and the latest evaluation of disease activity. By utilizing energy, insights can be gained into how a patient’s state may transit in the future and the optimal timing for effective treatment. We have added these sentences to the “Key results” section (page 27, line 417 to 420).

3) In the "Interpretation" section, the authors highlight the quantitative elucidation of significant treatment effects in terms of timing and duration. Please specify which particular result or findings you are referencing here.

Response:

Thank you for your valuable assessment. In our study, patient response to drug therapy and improved disease status showed a significant increase for about 6 months after the start of treatment and plateauing after 1 year. Furthermore, it was demonstrated that the patient population could be divided into three clusters: “toward good stability,” “toward poor stability,” and “unstable.” We have added these sentences to the “Interpretation” section (page 28, line 433 to 436)

4) Furthermore, in the "Interpretation" section, there is a suggestion that for certain patients who might be heading towards a treatment dead-end, a more potent treatment choice could be essential. Given that this study is anchored in a T2T-based approach, is it conceivable that such enhanced treatment decisions could be made?

Response:

Thank you for your valuable assessment. In the realm of AI applications in healthcare, there exists a fascinating parallel with the domain of chess, shogi, and go AI, where professionals occasionally find themselves defeated by seemingly unorthodox moves beyond human comprehension. Similarly, in the context of real-world practice visualization derived from our methodologies, instances may arise wherein the recommended treatment deviates significantly from established clinical guidelines. Consider a scenario where, despite a low disease activity, AI suggests an early and intensive administration of biologic agents. In such cases, it becomes imperative to hypothesize that the proposed therapeutic approach by AI holds merit, prompting the need for rigorous validation through randomized controlled trials (RCTs). Our research suggests the emergence of an innovative approach on the future landscape of medical research and practice, tentatively termed "AI-based RCTs," to systematically investigate and corroborate the efficacy of AI-suggested clinical interventions. We have added these sentences to the interpretation section (page 28, line 441 to page 29, line 452).

Minor comments

1) The units of measurement must be explicitly stated in the table for clarity.

Response:

We agree with your assessment. We have specified the units of measurement in the table1 and Table3. (page 17, line 298, page 24, line 370)

2) On Page 5, Line 50, the statement "The importance of early disease detection and treatment initiation." stands incomplete. Kindly provide context or rephrase for completeness.

Response:

We agree with your assessment. The difference between these two clusters is marginal; however, there is a possibility that the initiation of effective treatment was delayed in the “unstable” cluster due to the relatively favorable initial condition at the time of the first visit. Further research is warranted for subsequent investigations; however, the results suggest the importance of early disease detection and treatment initiation, especially in “unstable” cases. We have added these sentences to the key results section (page 26, line 407 to page 27, line 411)

Reviewer 2#:

The major problem of this manuscript is the definition of the terms "Stability" and "Remission". The authors should better to make sure the definitions and keep those terms consistently throughout the manuscript.

1) The term "remission" appears in this manuscript at several different situations and each implies a different meaning. For example, the word "remission" is used in "good stability leading to remission" and it used in each of the variables used in the energy landscape analysis. Additionally, the definition of "good stability" is "functional remission". From the above, it is necessary that the readers should guess the meaning of "remission" each time when it appears. I suggest that the authors consider improving on this terminology of “remission”.

2) In the methods section, the authors described that, “Good stability” was defined as meeting functional remission criteria based on the Health Assessment Questionnaire (HAQ), and “poor stability” was defined as dying or falling below the energy threshold without functional remission. This complicated definition makes it very difficult to understand Figure 5. According to the definition, G-L seems to be a combination of good stability = functionally good state and Low-energy = stability (+), which is still understandable. However, the most difficult one is the P-H combination, which is a combination of poor stability = functionally poor with falling below the threshold of energy (stability +) and High-energy = stability (-). This was quite confusing for the readers. I think that the "stability" is essentially defined from the result of high or low energy by energy landscape analysis.

Response:

Due to the strong correlation between 1) and 2), we have addressed them together in our response. We agree with your assessment. Thank you for the valuable suggestion. We have revised the definition of “poor stability” to be clarified based on the results of energy level using energy landscape analysis. (page 9, line 148 to line 151; page 18, line 310 to line 312; page 26, line 392 to line 393;).

3) The authors probably described Poor as Bad initially. I have found remnants of this in the text and in the Figure caption, Table. This made it very difficult to read, especially in the case of abbreviations such as B-L, as I could not keep up with their understanding. Please make sure to use the word "Poor" in a consistent manner.

Response:

Thank you for the valuable suggestion. We have reviewed and revised the figure captions and tables. (page 23, line 353 to line 355).

4) Please clarify 1) and 2) above and then rewrite Figure 1. This is the figure that confused me the most because the terms of “Good or Poor stability”, “remission” are mixed up in this figure. As a minor point, the word "Energy-igh" in the figure is probably a typo for -high, so please correct it.

Response:

We agree with your assessment. We have updated the Figure 1 (page 8, line 137).

---

## [Decision Letter · Decision Letter 1]

2 Apr 2024

Energy landscape analysis and time-series clustering analysis of patient state multistability related to rheumatoid arthritis drug treatment: the KURAMA cohort study

PONE-D-23-27887R1

Dear Dr. Yamamoto,

We’re pleased to inform you that your manuscript has been judged scientifically suitable for publication and will be formally accepted for publication once it meets all outstanding technical requirements.

Kind regards,

Masataka Kuwana, MD, PhD

Academic Editor

PLOS ONE

Additional Editor Comments (optional):

Reviewers' comments:

Reviewer's Responses to Questions

**Comments to the Author**

1. If the authors have adequately addressed your comments raised in a previous round of review and you feel that this manuscript is now acceptable for publication, you may indicate that here to bypass the “Comments to the Author” section, enter your conflict of interest statement in the “Confidential to Editor” section, and submit your "Accept" recommendation.

Reviewer #2: All comments have been addressed

2. Is the manuscript technically sound, and do the data support the conclusions?

Reviewer #2: Yes

3. Has the statistical analysis been performed appropriately and rigorously? 

Reviewer #2: Yes

4. Have the authors made all data underlying the findings in their manuscript fully available?

Reviewer #2: Yes

5. Is the manuscript presented in an intelligible fashion and written in standard English?

Reviewer #2: Yes

6. Review Comments to the Author

Reviewer #2: The authors well revised and answered our review comments. This version appeared better for publications. Findings are also important for future discussion.

7. PLOS authors have the option to publish the peer review history of their article (what does this mean?). If published, this will include your full peer review and any attached files.

Reviewer #2: **Yes: **Yasushi Kondo
